∂ | **Open Peer Review** | Human Microbiome | Research Article

# High-resolution metabolomic analysis of stool reveals expanded biomarkers of *C. difficile* colitis and insights into pathophysiology

Nirja Mehta,[1] Vincent Guzzetta,[2] Ken H. Liu,[3] Andrew S. Webster,[1] Max W. Adelman,[4,5,6,7] Eric C. Fitts,[8] Dean P. Jones,[3] Colleen S. Kraft,[1,8] Michael H. Woodworth,[1] Jeffrey M. Collins[1]

**ABSTRACT** *Clostridioides difficile* infection (CDI) is facilitated by gut microbiome disruption and associated metabolic disturbances. While many prior studies have focused on the microbial composition of CDI stool, fewer have explored the fecal metabolome using untargeted high-resolution metabolomics (HRM). To characterize the metabolic phenotype of CDI, we performed untargeted HRM on stool samples from 55 CDI patients and 72 healthy controls. CDI was associated with marked alterations in stool metabolites, mapping to 14 significantly dysregulated pathways, including tryptophan metabolism, bile acid metabolism, short-chain fatty acid metabolism, and fatty acid oxidation. These findings suggest that CDI is associated with a distinct fecal metabolic milieu that may promote inflammation, impair colonization resistance, and facilitate *C. difficile* replication. Our results corroborate previous studies and support further investigation of the role of microbial and immune signaling in CDI as well as metabolic biomarkers and therapeutic targets.

**IMPORTANCE** As we learn more about the essential role of gut microbes in human health and disease, there is an effort to identify and characterize stool biomarkers that distinguish patients with microbiome-associated disease from healthy individuals. *Clostridioides difficile* infection contributes to excess healthcare burden and is tightly linked to disruption of the gut microbiome (usually by use of antimicrobials). We compared the metabolome of healthy stool and *C. difficile*-infected stool to identify metabolites that correlate with disease and may support the development of improved diagnostic tests for *C. difficile* infection. In comparing these two groups, we found chemical pathways that strongly correlate with disease and others more associated with healthy stool, including bile acid biosynthesis, tryptophan metabolism, and carnitine activation.

**KEYWORDS** *C. difficile*, *Clostridioides*, diagnostic testing, metabolomics

*C*lostridioides difficile infection (CDI) presents an extensive financial and operational burden to healthcare systems, including $5 billion in annual healthcare costs and 30,000 patient deaths annually (1, 2). CDI is enabled by perturbations in the gut microbiome. Many studies evaluate microbial populations in the setting of CDI through metagenomics, but fewer studies have evaluated the gut metabolome using untargeted metabolomics. The function of gut microbial populations can be elucidated by the study of the metabolome. Microbial populations produce metabolites that can facilitate or inhibit *C. difficile* replication, toxin production, and evasion of the host immune system (3–6).

Evaluating the gut metabolome of patients with rCDI and healthy controls can uncover key microbial pathways that facilitate CDI. Prior work has demonstrated the role of bile acid metabolism in facilitating *C. difficile* germination, short-chain fatty acid (SCFA)

Address correspondence to Michael H. Woodworth, mwoodwo@emory.edu.

C.S.K. is a consultant for Ferring Pharmaceuticals and a scientific advisor for Seres Therapeutics. The other authors do not report any conflict of interest.

metabolism in mediating colonic inflammation, and the metabolism of amino acids for both nutrition and influencing host immune function (4–6). A greater understanding of these processes may help identify vulnerable microbiomes among high-risk patients.

With high-resolution metabolomics (HRM), it is possible to simultaneously measure thousands of small molecules in stool samples (7, 8). We applied an untargeted HRM approach to characterize the differences in the stool metabolome between healthy subjects and those with CDI. Our analysis found a clear divergence in the stool metabolome of CDI patients versus the stool of healthy persons. It also implicates a set of chemical pathways that may be important in CDI pathophysiology (9–15).

## RESULTS

We collected stool specimens from 55 patients with CDI and 72 healthy donor subjects between 2019 and 2021. CDI diagnosis was defined as *C. difficile* NAAT-positive cases with associated diarrhea treated with anti-CDI antibiotics. CDI patients were recruited from both inpatient and outpatient settings, while control stool consisted of de-identified specimens obtained from healthy donor cohorts (Table 1). CDI patients were older (median age: 61 versus 26 years) and had a higher body mass index (BMI) (median: 26.9 versus 21.8 kg/m$^2$) than controls. CDI patients were 53% female, whereas the control population was approximately 68% female. In 69% of cases, CDI patients had both positive *C. difficile* NAAT and toxin EIA testing. The majority (37/55, 67%) of CDI patients had no history of a prior CDI episode in the prior 12 months. Most CDI patients (46/55, 84%) were hospitalized at the time of diagnosis (Table 2).

To evaluate the stool metabolic phenotypes associated with CDI, we performed untargeted metabolomics on stool samples using dual hydrophilic interaction liquid chromatography (HILIC) and C18 liquid chromatography in positive and negative ionization modes, respectively (see Materials and Methods for full details), to maximize metabolite detection across chemical classes. After controlling for age, sex, and BMI, marked differences were apparent in the stool metabolome of patients with CDI compared to controls. Using a false discovery rate (FDR)-corrected *P* value of 0.05, 239 metabolite features were significantly increased in the stool of CDI participants in HILIC positive mode, while 1,924 features were significantly decreased (Fig. 1A). Similarly, in C18 negative mode, 119 metabolite features were significantly increased in the stool of CDI patients versus controls, while 821 were significantly decreased (Fig. 1B). Utilizing the mummichog pathway analysis program, we found these metabolites mapped to 14 stool metabolic pathways that were significantly different in CDI patients versus controls (Fig. 1C). We found metabolites significantly upregulated in the stool of patients with CDI mapped to the carnitine shuttle pathway. Conversely, pathways in leukotriene, steroid, arachidonic acid, and prostaglandin metabolism were downregulated in CDI stool, as was tyrosine and tryptophan metabolism.

Within these pathways, we then examined individual metabolites with high-confidence annotations to see how they were either disrupted or enriched in CDI stool. We first examined metabolites in the tryptophan metabolic pathway, which performs important immune regulatory functions for gut homeostasis. (16, 17). A two-way

TABLE 1 Demographics of patients with *C. difficile* infection and controls

| | *C. difficile* (*N* = 55) | Control (*N* = 72) | Overall (*N* = 127) |
|---|---|---|---|
| Gender | | | |
| Female | 29 (52.7%) | 49 (68.1%) | 78 (61.4%) |
| Male | 26 (47.3%) | 23 (31.9%) | 49 (38.6%) |
| Age | | | |
| Median [min, max] | 61.0 [31.0, 89.0] | 26.0 [18.0, 65.0] | 36.0 [18.0, 89.0] |
| Missing | 0 (0%) | 1 (1.4%) | 1 (0.8%) |
| BMI_num | | | |
| Median [min, max] | 26.9 [15.0, 48.3] | 21.8 [18.4, 24.6] | 23.4 [15.0, 48.3] |
| Missing | 2 (3.6%) | 15 (20.8%) | 17 (13.4%) |

**TABLE 2** Clinical characteristics of *C. difficile* infection cases

| Characteristic | $N = 55^a$ |
| --- | --- |
| EIA positive | 38 (69%) |
| Prior episodes[b] | |
| 0 | 37 (67%) |
| 1 | 15 (27%) |
| 2 | 1 (1.8%) |
| 3 | 1 (1.8%) |
| 4 | 1 (1.8%) |
| Immunocompromised[c] | 17 (31%) |
| Inpatient | 46 (84%) |
| Recent antibiotic exposure[d] | |
| Cephalosporin | 22 (40%) |
| Macrolide | 4 (7%) |
| Trimethoprim-sulfamethoxazole | 4 (7%) |
| IV vancomycin | 11 (20%) |
| Ampicillin-sulbactam/piperacillin-tazobactam | 7 (13%) |
| Metronidazole | 5 (9%) |
| Carbapenem | 4 (7%) |
| Fluoroquinolone | 9 (16%) |
| Linezolid | 2 (4%) |
| Other | 4 (7%) |
| No known recent antibiotics | 15 (27%) |

[a]n, (%).
[b]Within 365 days.
[c]Defined as receipt of a solid organ transplantation, active hematologic malignancy, daily use of the equivalent of >20 mg prednisone.
[d]Antibiotics in 4 weeks prior to CDI (many patients with multiple antibiotic class exposures).

hierarchical clustering analysis of annotated metabolites from the tryptophan pathway demonstrated strong separation between persons with and without CDI (Fig. 2A). Notably, stool from participants with CDI that was toxin-positive did not cluster separately from those with stool that was toxin-negative. CDI stool had significantly decreased concentrations of multiple tryptophan metabolites, including 2-oxoadipate ($P < 0.0001$; Fig. 2B), 3-hydroxyanthranilate ($P < 0.0001$; Fig. 2C), and anthranilate ($P = 0.004$; Fig. 2D). This indicates that the stool metabolic environment in CDI is depleted of tryptophan metabolites, potentially abrogating a host response that limits inflammation.

We also examined metabolites mapping to polyunsaturated fatty acid metabolism and eicosanoid pathways, which include lipid mediators that regulate inflammation: arachidonic acid, prostaglandin, and leukotriene metabolism, as well as neuroprostane formation and 3-oxo-10R-octadecatrienoate beta-oxidation. We again found near-complete separation between CDI stool and control stool using two-way hierarchical clustering of metabolites mapping to these pathways (Fig. 3A). These metabolites are often upregulated in inflammation, but interestingly, CDI stool had significantly lower concentrations of nearly all metabolites in these pathways, including arachidonic acid ($P = 0.0002$; Fig. 3B), leukotriene A4 ($P < 0.0001$; Fig. 3C), and 12-HHT ($P = 0.0002$; Fig. 3D). Together, these indicate CDI stool in humans is an environment characterized by depletion of inflammatory fatty acid signaling.

Finally, we examined the carnitine shuttle and bile acid pathways due to their association with energy metabolism, regulation of inflammation, and increased *C. difficile* germination (18). We found that stool from CDI patients had increased concentrations of long-chain acylcarnitines, consistent with an environment of impaired fatty acid oxidation (Fig. 4A through C). With regard to bile acids, we found that bile acid precursors, including 3-beta-7-alpha-dihydroxy-5-cholestenoate, were generally depleted from CDI stool ($P < 0.0001$; Fig. 4D), while the conjugated bile acids taurodeoxycholate and glycocholate were significantly elevated in CDI stool versus control stool ($P = 0.03$ and

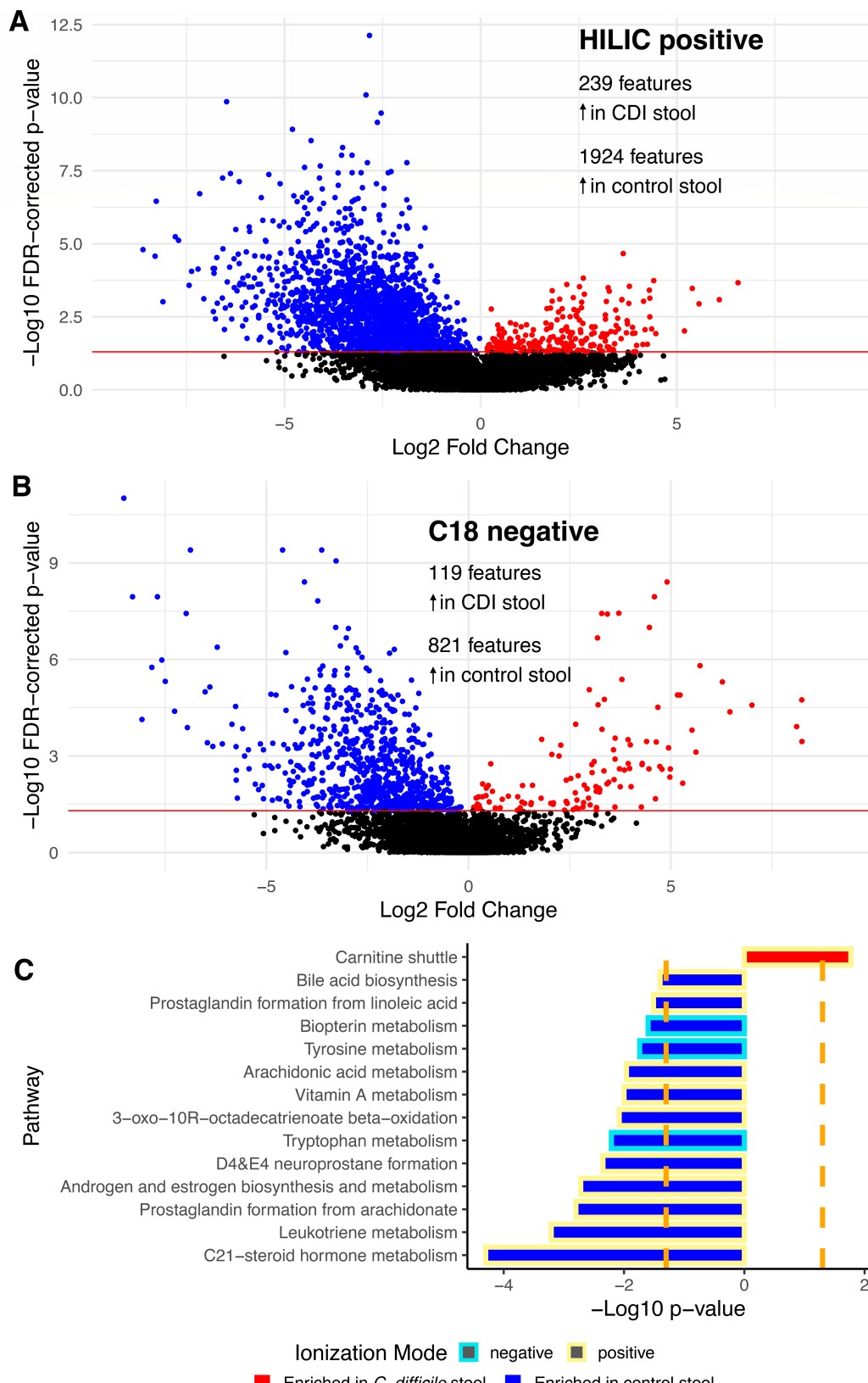

FIG 1   (A) Volcano plots of stool metabolic features significantly increased in *C. difficile* infection stool versus control stool (red; FDR-corrected *P* value value <0.05) and those significantly decreased (blue) using positive ionization mode with HILIC chromatography and (B) negative ionization mode with C18 chromatography. (C) Metabolic pathways significantly increased in *C. difficile* infection stool versus control stool (red) and those that significantly decreased (blue; *P* < 0.05).

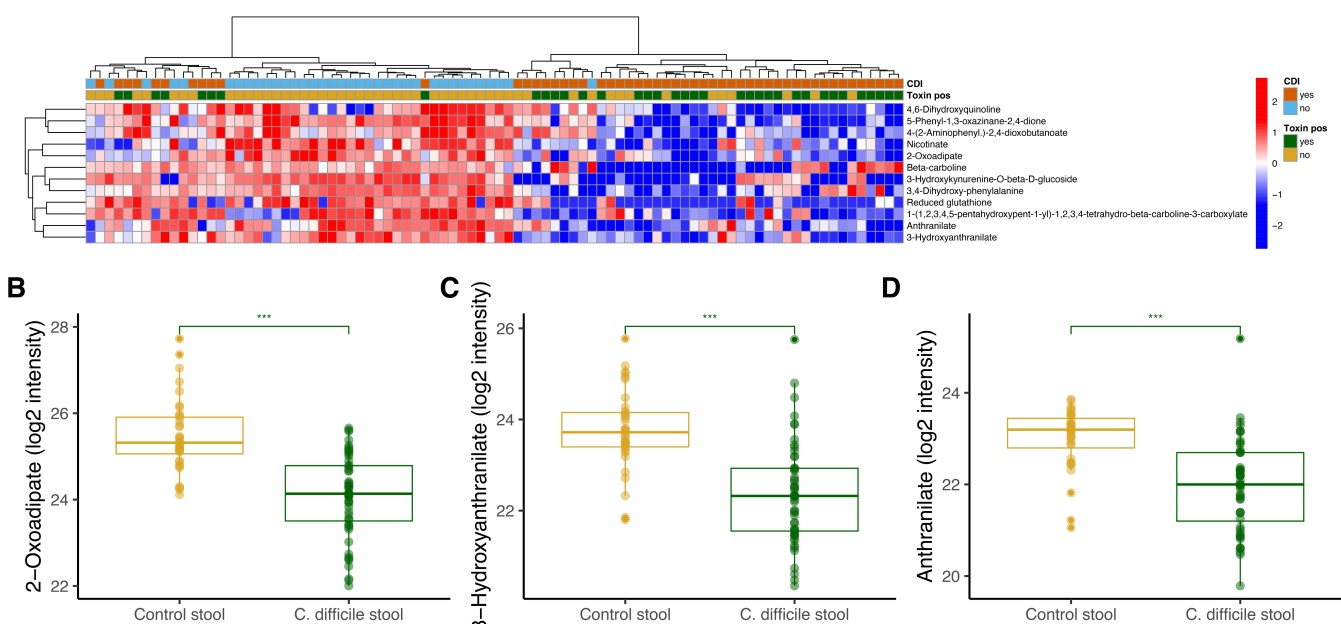

**FIG 2** (A) Two-way hierarchical clustering analysis of all metabolites mapping to the tryptophan pathway annotated by their *C. difficile* infection status (positive, orange and negative, light blue) as well as stool toxin assay results (green, positive and gold, negative). Log2 intensity measure of stool (B) 2-oxoadipate, (C) 3-hydroxyanthranilate, and (D) anthranilate in participants with *C. difficile* infection versus controls. The box plots depict the minimum and maximum values (whiskers) and the median. The length of the box represents the interquartile range. A linear regression model controlling for age and sex was used to compare metabolite levels between groups (***$P < 0.001$).

0.002, respectively; Fig. 4E and F). Our results suggest that CDI is associated with a stool metabolic milieu characterized by impaired oxidation of long-chain fatty acids, decreased concentrations of bile acid precursors, and increased concentrations of conjugated primary bile acids, all of which have been shown to enhance *C. difficile* germination (18).

## DISCUSSION

It is currently understood that the stool of CDI patients differs widely in microbial populations and volatile compound content when compared to stool from healthy persons (19–21). Our study builds on this understanding by investigating the fecal metabolic environment in CDI patients, comparing it to that of healthy controls. By linking these metabolic differences to known microbial pathways, we underscore their potential significance in the pathophysiology of CDI. Our analysis demonstrated differences in several metabolic pathways previously observed in human studies, including tryptophan metabolism, carnitine metabolism, bile acid metabolism, and SCFA production (4, 6, 22).

Tryptophan metabolism emerged as one of the distinguishing features of CDI stool. We observed an increase in free tryptophan, accompanied by lower levels of downstream metabolites, such as anthranilate, 3-hydroxyanthranilate, and kynurenine, in CDI patients. This aligns with prior studies, including McMillan et al., which found a similar enrichment of tryptophan metabolites in CDI patients prior to fecal microbiota transplantation (4). In a study of patients with CDI and other causes of hospital-acquired diarrhea, there was a signal toward a difference in tryptophan catabolism among patients with CDI compared to non-CDI patients (5).

Tryptophan catabolism is facilitated by gut microbiota. Germ-free mice are known to have higher levels of free tryptophan in the gut, and antibiotic-induced disruptions to the microbiome similarly elevate tryptophan concentrations (23–25). In murine models,

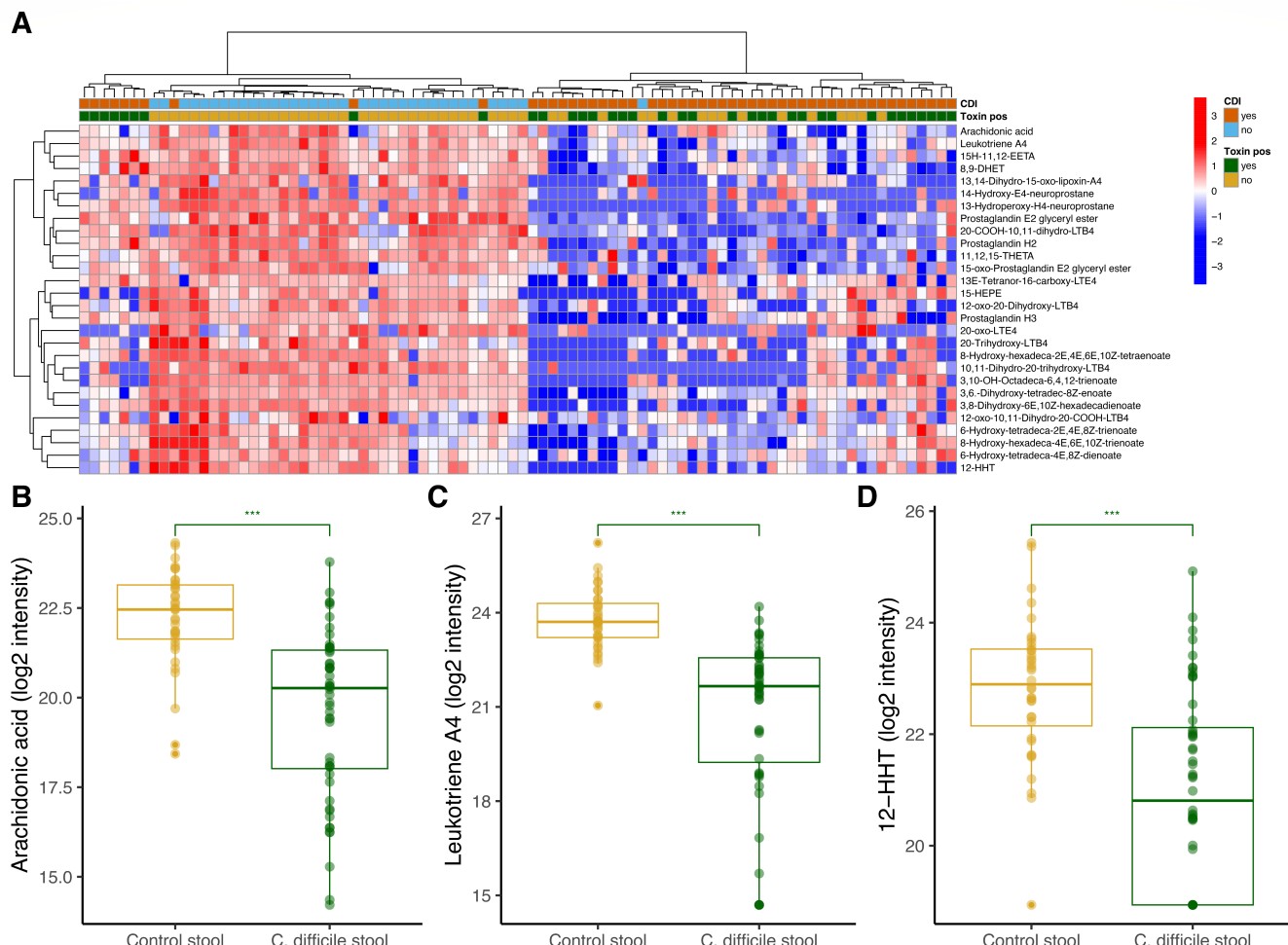

**FIG 3** (A) Two-way hierarchical clustering analysis of all annotated metabolites mapping to prostaglandin, leukotriene, and arachidonic acid metabolic pathways. Each column represents a participant annotated by their *C. difficile* infection status (positive, orange and negative, light blue) as well as stool toxin assay results (green, positive and gold, negative). Log2 intensity measure of stool (B) arachidonic acid, (C) leukotriene A4, and (D) 12-HHT in participants with *C. difficile* infection versus controls. The box plots depict the minimum and maximum values (whiskers) and the median. The length of the box represents the interquartile range. A linear regression model controlling for age and sex was used to compare metabolite levels between groups (***$P < 0.001$).

tryptophan conversion to kynurenine is inhibited during *C. difficile* colitis, which limits modulation of the inflammatory response facilitated by kynurenine and other downstream metabolites (26). Kynurenines limit pro-inflammatory host responses through promotion of regulatory T-cell differentiation, as well as limit neutrophil accumulation in the lamina propria (27). Limiting these anti-inflammatory signals in the stool metabolome may help *C. difficile* promote an inflammatory environment where it has a survival advantage.

Contrary to expectations based on animal models, our stool metabolomic analysis displayed lower stool concentrations of pro-inflammatory mediators from the eicosanoid pathway among CDI patients. For instance, there were lower levels of arachidonic acid, as well as the pro-inflammatory arachidonic derivatives leukotriene A4 and 9-HETE. This differs from animal models, which demonstrate that C. *difficile* Toxin A can potentiate the natural production of leukotriene B4 produced from leukotriene A4 (28, 29). However, a study by Trindade et al. demonstrated that abolishing leukotriene signaling had no effect on the inflammatory response and intestinal damage of CDI in their animal model: both neutrophilic cytokine bursts and anti-toxin immunoglobulin production continued independent of leukotriene signaling (30). Similarly, a clinical trial of a developed LTB4 inhibitor in humans had no effect on the clinical severity of inflammation in ulcerative

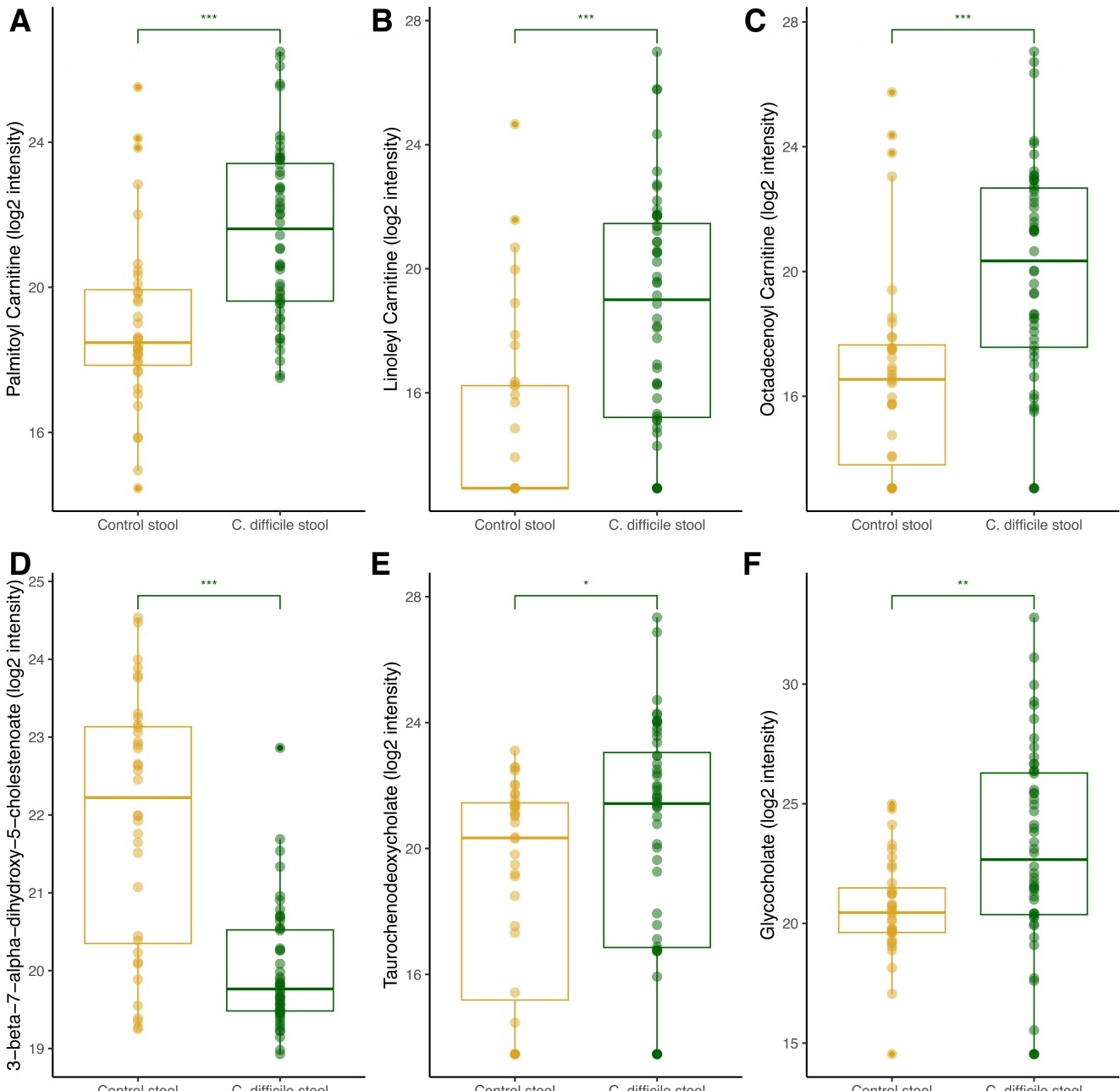

**FIG 4** Comparison of acylcarnitine and bile acid metabolites comparing *C. difficile* infection versus controls. Log2 intensity measure of (A) palmitoyl carnitine, (B) linoleyl carnitine, (C) octadecenoyl carnitine, (D) 3-beta-7-alpha-dihydroxy-5-cholestenoate, (E) taurochenodeoxycholate, and (F) glycocholate. The box plots depict the minimum and maximum values (whiskers) and the median. The length of the box represents the interquartile range. A linear regression model controlling for age and sex was used to compare metabolite levels between groups (*$P < 0.05$, **$P < 0.01$, and ***$P < 0.001$).

colitis (31). Thus, it is possible that these metabolites are not primary contributors to the pro-inflammatory state in CDI.

These metabolites may also be important for *C. difficile* colonization resistance, and their depletion could indicate an environment of higher CDI risk. Arachidonic acid has inhibitory effects on Enterobacterales, which are more common in the gut microbiome of CDI patients (32). There some data that higher baseline levels of arachidonic acid in the gut lumen limit microbes that support *C. difficile* colonization (33). Additionally, there is evidence that some prostaglandins, which are also derived from arachidonic acid, may play an inhibitory role in colitis (34).

Metabolites from the carnitine shuttle pathway were enriched in *C. difficile* stool. Carnitine metabolism is facilitated by many Enterobacteriaceae, which are typically more abundant in CDI patients (35). McMillan et al. demonstrated a significant decrease in acylcarnitines in CDI patients following fecal microbiota transplantation and restoration of the microbiome (4). Mutations in carnitine synthesis have been shown to play a role in IBD, and L-carnitine treatment in animal models has demonstrated reduction of inflammatory phenotypes (36). Both of these observations suggest that increases in fatty acid oxidation are essential to restore immune homeostasis.

Stool metabolism of SCFAs was also impaired in CDI patients versus healthy controls. For example, butanoic acid was significantly increased in CDI stool. This finding is consistent with other human metabolomics studies, which showed patients with CDI had a significantly higher abundance of butyrate in the stool than patients with diarrhea of other causes (5). *In vitro*, SCFAs have been shown to increase toxin production (37), so their accumulation may further improve the environmental niche for *C. difficile* (5).

Bile acids play a key protective role in *C. difficile* colonization resistance. Unconjugated primary bile acids (cholic acid and chenodeoxycholic acid) are conjugated with taurine or glycine in the liver and traverse to the gut as primary bile acids (38). Cholate-based primary bile acids trigger *C. difficile* germination (39, 40). Commensal gut microbiota, including Bacteroidetes, Firmicutes, and Actinobacteria, produce bile acid hydrolases which hydrolyze conjugated primary bile acids to secondary bile acids and inhibit germination. We found that the conjugated primary bile acids taurochenodeoxycholate and glycocholate were significantly increased in CDI stool, suggesting reduced deconjugation by the intestinal microbiota. Conversely, bile acid precursors were significantly decreased. This finding has been replicated in several human studies. In a study of 53 patients with recurrent CDI, following therapy for the index episode, patients who had higher abundance of primary bile acids in stool also had slower recovery of microbial diversity and went on to have subsequent episodes of CDI (19). Patients who receive microbiome therapeutics have a more rapid recovery of secondary bile acids (4, 41). These findings indicate that accumulation of conjugated primary bile acids provides a metabolic milieu that supports *C. difficile* colonization and CDI pathogenesis (4, 19, 41). Interestingly, we did not observe the same alterations in Stickland metabolites, such as products of proline and leucine reduction, that have been reported as discriminatory in other studies (4, 6, 22).

Control stool was enriched for C21-steroid hormone metabolism and androgen and estrogen biosynthesis; this is likely reflective of the pro-inflammatory state of the CDI gut (42). These hormones, especially androgens, have an anti-inflammatory effect on the gut (43). Increased concentration of sex hormones has been seen in increased microbial diversity in both men and women (44). Antibiotic treatment, more commonly seen among CDI patients, is noted to disrupt steroid hormone metabolism (45). Prior data demonstrate that patients who experienced recurrent CDI had a trend toward corticosteroid metabolism in the gut when compared to patients who did not, which denotes a possible protective role of corticosteroid metabolism in preventing CDI pathogenesis (19).

This study has notable limitations. Patients with CDI were compared to healthy controls. The CDI group was heterogeneous, including hospitalized patients, outpatients, and patients with various comorbid conditions and taking a variety of prescription medications, which may contribute to differences in the gut metabolome when compared to healthy, relatively young controls outside of CDI.

However, despite these differences in the study groups, our study replicates the effects seen in prior studies, which may indicate the consistency of these metabolites in facilitating CDI.

Future studies should longitudinally evaluate changes in the metabolome during CDI and after resolution of diarrhea to evaluate biomarkers comparing restoration of infection. Comparing the metabolomes of patients with infection and colonization, as

determined by NAAT testing and symptoms, could elucidate candidates for diagnostic biomarkers.

This study helps to characterize the stool metabolic phenotypes associated with CDI in humans. Our findings support key roles for bile acid metabolism, SCFA metabolism, and tryptophan metabolism in enhancing CDI pathogenesis. We further show that CDI is associated with impaired fatty acid oxidation and depletion of inflammatory lipid mediators. Improved understanding of how these metabolic pathways enhance CDI risk and/or pathogenesis could lead to host-directed therapies that enhance colonization resistance and CDI virulence.

## MATERIALS AND METHODS

### Study population

Stool from CDI patients was collected from adult patients at the time of CDI diagnosis from the Emory University Medical laboratory between June and December of 2021. CDI diagnosis was defined as a positive *C. difficile* toxin NAAT result coupled with clinical treatment for CDI on chart review. Stool samples collected from both inpatient and outpatient collections were included. A chart review was conducted on each CDI patient to obtain demographic and clinical characteristics. Recurrent CDI was defined as a prior CDI diagnosis within the past year. The control group consisted of healthy adults between the ages of 18–40. The inclusion criteria for this group were kept rigorous to define a disease-free population based on age, weight, diet, lifestyle, routine laboratory tests, and medical history.

### Stool sample collection and processing

Stool samples were collected at Emory hospital laboratory from both inpatient and ambulatory patients with CDI, as well as from two healthy cohorts from the Emory Vaccine Center and Microbiome Enrichment Program. Stool samples were placed on ice immediately after collection and stored at 4°C for up to 24 h. After aliquoting, they were moved to a −80°C freezer for long-term storage. The stool processing protocol was adapted from published methods (9). In brief, 100 mg of stool from stool donors was thawed and dissolved in 400 µL of PBS on ice and vortexed for 30 s. It was then centrifuged for 10 min at 10K rpm at 4°C. Fifty microliters of supernatant was combined with 100 µL of acetonitrile and centrifuged at 16K rpm for 10 min at 4°C and transferred to an autosampler tube. This process was repeated for PBS, where one set of samples was subjected to sonication in a bath sonicator at 40 Hz for 10 min at 4°C.

### High-resolution metabolomics analysis

Samples were de-identified prior to transfer to the analytical laboratory, where personnel were blinded to clinical and demographic data. Five microliter aliquots of sample extracts were analyzed using liquid chromatography and high-resolution mass spectrometry (Dionex Ultimate 3000, HF Q-Exactive, Thermo Scientific). Three technical replicates were performed for each sample to ensure data quality and reproducibility. Sample extracts were injected and analyzed using dual HILIC positive and also reverse phase (C18) negative chromatography. Analyte separation for HILIC was performed with a Waters XBridge BEH Amide XP HILIC column (2.1 × 50 mm$^2$, 2.6 µm particle size) and gradient elution with mobile phases A: LCMS grade water, B: LCMS grade acetonitrile, and C: 2% formic acid. The initial 1.5 min period consisted of 22.5% A, 75% B, and 2.5% C, followed by a linear increase to 75% A, 22.5% B, and 2.5% C at 4 min and a final hold of 1 min. C18 chromatography was performed on an end-capped C18 column (Higgins Targa C18 2.1 × 50 mm$^2$, 3 µm particle size) with mobile phases A: water, B: acetonitrile, and C: 10 mM ammonium acetate. The initial 1 min period consisted of 60% A, 35% B, and 5% C, followed by a linear increase to 0% A, 95% B, and 5% C at 3 min and held for the remaining 2 min. For both methods, the mobile phase flow rate was 0.35 mL/min for

the first min and increased to 0.4 mL/min for the final 4 min. The HRMS was operated at 120K resolution, and MS1 spectra were collected from 85 to 1,275 $m/z$. Tune parameters for sheath gas were 45 for ESI+ and 30 for ESI−. Auxiliary gas was set at 25 for ESI+ and 5 for ESI−. Spray voltage was set at 3.5 kV for ESI+ and −3.0 kV for ESI−. Data were extracted and aligned using apLCMS (10) and xMSanalyzer (11) with each feature defined by specific $m/z$ value, retention time, and integrated ion intensity (12). The intensity values for each feature were summarized by the median across technical replicates (13). When multiple adducts were detected, the most reproducible MS1 adduct was selected for analysis. Metabolite annotations had to meet three criteria: (i) a high-confidence mass match to either the M + H or M − H adduct, (ii) membership in a significantly regulated metabolic pathway, and (iii) correlation with other annotated metabolites in a significant metabolic pathway. When possible, identities were further supported as either medium-confidence or high-confidence using xMSAnnotator, a multi-stage clustering algorithm to derive compound annotation and confidence scores (7).

## Data analysis

Statistical comparisons of metabolite intensity values (abundance) were performed in R version 4.2.2. Metabolite intensity values were log2-transformed and quantile-normalized. Normalized metabolite intensities from CDI stool were compared to those from control stool using linear regression, controlling for age, BMI, and sex. BMI was imputed in one CDI case. Differences between groups were adjusted for multiple comparisons using the Benjamini-Hochberg FDR, and those metabolic features with an FDR < 0.05 were considered statistically significant (14). Significant features, along with their $m/z$ values, retention times, and FDR-corrected $P$ values, were then provided as input to mummichog, a Python package for pathway and network enrichment analysis in metabolomics that does not require prior metabolite identification. (https://github.com/shuzhao-li/mummichog) (15). Mummichog maps significant features (the top 25% of features with FDR-corrected $P$ values <0.05 in the null distribution model) to all possible candidate metabolites and projects them onto known metabolic networks, such as those from the KEGG database (46). It then evaluates whether clusters of mapped metabolites are enriched in specific pathways compared to random expectation. Features mapping to significantly enriched clusters are considered more likely to represent biologically relevant metabolites. A pathway overlap of at least five metabolites with a $P < 0.05$ was used to determine significant pathways. Two-way hierarchical clustering analysis was performed using the pheatmap package in R (Kolde R. 2025. pheatmap: Pretty Heatmaps. R package version 1.0.13, https://github.com/raivokolde/pheatmap), while all other data visualizations were performed using ggplot2 (47).

## ACKNOWLEDGMENTS

Research reported in this publication was supported by the Emory Prevention Epicenter Program (PEACH) through the Centers for Disease Control and Prevention (CDC) (U54CK000601 to M.H.W.), as well as the National Institute for Allergy and Infectious Diseases (K23AI144036 to M.H.W.). Additional support was provided by the National Institute of Allergy and Infectious Diseases (NIAID) grant K23 AI144040 to J.M.C., NIH National Center for Advancing Translational Science grant UL1 TR002378 to J.M.C., National Institute of Allergy and Infectious Diseases at the National Institutes of Health (grant number K23AI185174 to M.W.A.), and the Houston Methodist Academic Institute (Clinical Scholars Award to M.W.A.).

## AUTHOR AFFILIATIONS

[1]Division of Infectious Diseases, Department of Medicine, Emory University School of Medicine, Atlanta, Georgia, USA
[2]Department of Medicine, Emory University School of Medicine, Atlanta, Georgia, USA

[3]Division of Pulmonary, Allergy and Critical Care Medicine, Department of Medicine, Emory University School of Medicine, Atlanta, Georgia, USA

[4]Division of Infectious Diseases, Department of Medicine, Houston Methodist Hospital, Houston, Texas, USA

[5]Center for Infectious Diseases, Houston Methodist Research Institute, Houston, Texas, USA

[6]Division of Pulmonary, Critical Care, and Sleep Medicine, Houston Methodist Hospital, Houston, Texas, USA

[7]Department of Medicine, Weill Cornell Medical College, New York, New York, USA

[8]Department of Pathology, Emory University School of Medicine, Atlanta, Georgia, USA

## AUTHOR ORCIDs

Nirja Mehta  https://orcid.org/0000-0001-6324-0077
Ken H. Liu  http://orcid.org/0000-0002-9736-5828
Max W. Adelman  http://orcid.org/0000-0002-9277-6046
Colleen S. Kraft  http://orcid.org/0000-0003-1757-8477
Michael H. Woodworth  http://orcid.org/0000-0002-6181-4599
Jeffrey M. Collins  http://orcid.org/0000-0002-6574-0873

## AUTHOR CONTRIBUTIONS

Nirja Mehta, Writing – original draft, Writing – review and editing | Vincent Guzzetta, Writing – original draft | Ken H. Liu, Investigation, Methodology | Andrew S. Webster, Resources | Max W. Adelman, Writing – review and editing | Eric C. Fitts, Resources | Dean P. Jones, Formal analysis | Colleen S. Kraft, Resources, Supervision | Michael H. Woodworth, Conceptualization, Validation | Jeffrey M. Collins, Conceptualization, Supervision

## ETHICS APPROVAL

All patients consented to stool collection and analysis. Stool collection protocols were approved by the Emory University Institutional Review Board.

## ADDITIONAL FILES

The following material is available online.

Open Peer Review

**PEER REVIEW HISTORY (review-history.pdf).** An accounting of the reviewer comments and feedback.

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
