## [Reviewer comments · Microbiology Spectrum]

Microbiology Spectrum

High-resolution metabolomic analysis of stool reveals expanded biomarkers of *C. difficile* colitis and insights into pathophysiology

Nirja Mehta, Vincent Guzzetta, Ken Liu, Andrew Webster, Max Adelman, Eric Fitts, Dean Jones, Colleen Kraft, Michael Woodworth, and Jeffrey Collins

Corresponding Author(s): Nirja Mehta, Emory University

Review Timeline:

Submission Date:	November 10, 2025
Editorial Decision:	November 17, 2025
Revision Received:	March 17, 2026
Accepted:	March 20, 2026

Editor: Jan Claesen

Reviewer(s): The reviewers have opted to remain anonymous.

Transaction Report:

DOI: <https://doi.org/10.1128/spectrum.02826-25>

Re: Spectrum02826-25 (High-resolution metabolomic analysis of stool reveals expanded biomarkers of *C. difficile* colitis and insights into pathophysiology)

Dear Dr. Nirja Mehta:

Thank you for the privilege of reviewing your work. Below you will find my comments, instructions from the Spectrum editorial office, and the reviewer comments.

Thanks for addressing the Reviewer comments of your prior submission to a different ASM journal. I am pleased to inform you that your manuscript has been editorially accepted for publication. However, there are a few additional, Spectrum-specific questions in the submission form that need to be answered before the final decision. Once these are completed, please return your submission so that I can move your paper forward to acceptance.

Revision Guidelines

Sincerely,
Jan Claesen
Editor
Microbiology Spectrum

Reviewer Responses

Title High-resolution metabolomic analysis of stool reveals expanded biomarkers of *C. difficile* colitis and insights into pathophysiology

Authors

Nirja Mehta, Vincent Guzzetta, Ken Liu, Andrew Webster, Max W. Adelman, Eric Fitts, Dean Jones, Colleen S. Kraft, Michael H. Woodworth, Jeffrey M. Collins

Reviewer #1 (Comments to the Author (Required)):

The paper by Mehta et al. uses high resolution metabolomics to define the biomarkers in patients with CDI (n=55) and without CDI or healthy controls (n=72). They find that the stool metabolome of CDI patients has 14 altered pathways, including metabolism of tryptophan, bile acids, SCFAs, and fatty acid oxidation. This is a very simplistic conclusion based on a large and complex data set. It is not clear how the authors arrived at this result and how rigorous the statistical approaches were when looking at this patient population. The concept of the paper is of importance and there are other studies out there that have done this, which are cited. There seem to be many details that the authors skimmed over in the methods section making it hard to evaluate the main conclusions in the paper at this time. Major comments are below.

Major Issues:

1. Table 2 shows the clinical characteristics of the patients with CDI included in this study. There are patients with comorbidities, multiple relapses etc that are all lumped together into one group. These patients I am sure are also on antibiotics and other drugs.

Thank you for this comment. We have included antibiotic classes in the past 4 weeks in Table 2. We have also provided more details as a misc file to the journal.

All of these things need to be addressed when looking at this cohort compared to the healthy controls. I do not see details regarding this in the methods. I do see this listed in the discussion section as a limitation, but things like what antibiotics the person is on and what other issues they have would be important to decipher in this patient population.

We agree with the reviewer that participants with *C. difficile* infection exhibited heterogeneity regarding several clinical factors including antibiotic exposures prior to CDI onset, concomitant immunosuppression, and a history of prior CDI. In our clinical experience, this reflects the real-world diversity patients with CDI. We suggest that identifying robust metabolic signals in CDI stool specimens across this clinical background versus healthy donor stool enhances the value of this study. We have included additional clinical variables in the revised Table 2 as well as an additional table with individual patient characteristics to the journal.

There is a wealth of information missing from the methods section including the packages used for figure generation and analysis, FDR methods used, open source data is missing, along with the list of metabolites found in all figures. I would suggest the authors put together a github page, so the reader can see how they make the figures and raw data as well.

We agree clarification on this point is important. All figures were created in R using ggplot2, which is now cited in the revised methods. We used the Benjamini-Hochberg method to calculate the FDR, which is also cited in this resubmitted manuscript. [Line 285-301]

We have added the full name of the method to the text of the revised methods section. We have provided raw data used for figure generation.

2. It looks like the data was evaluated based on sex and age, but antibiotic usage and other co morbidities were not, which is pretty important. See comment above as well. Why did the authors not look at this too, in addition to age and sex?

Data was evaluated for age, sex and BMI. Given the variety of comorbid conditions and variety of antibiotics used, controlling for these variables was not thought to be practical. Instead the interest was to evaluate robust signals across a large population of patients with CDI. . However, there is a file of clinical characteristics under misc file.

3. Please provide more details in the methods on who patients were deemed CDI positive and rCDI patient status. Please include IRB and ethics information in the methods. There are very few details in this section.

These data were obtained from two different protocols. Both were approved by the Emory IRB. We have added this information to the methods section [Line 234-236]

4. Lack of details and open source information on metabolites, how they were identified, and what were the detailed the statistical methods used?

Metabolic features (i.e. unannotated metabolites) were log₂ transformed, quantile normalized, and compared between groups using linear regression controlling for age, sex, and BMI. FDR-corrected p-values of less than 0.05 were considered statistically significant. Pathway analysis was conducted using mummichog, a python-based package designed to determine metabolic pathway activity from high throughput, untargeted metabolomics data. The full details of the mummichog algorithm are published elsewhere (Li et al PLoS Computational Biology 2013) and the program is publicly available through github and Metaboanalyst. We have included additional details about this method as well as a link to the github page in the revised manuscript. [Line 282-301]

Metabolite annotations had to meet 3 criteria: 1) a high-confidence mass match to either the M+H or M-H adduct, 2) membership in a significantly regulated metabolic pathway, and 3) correlation with other annotated metabolites in a significant metabolic pathway. When possible, identities were further supported as either medium- or high-confidence using xMSAnnotator, a multi-stage clustering algorithm to derive compound annotation and confidence scores (Uppal et al. Anal Chem 2017). These details have been added to the revised methods section. [Line 272-279]

Reviewer #2 (Comments to the Author (Required)):

In this manuscript, the authors performed metabolomic analyses on stool samples from Clostridioides difficile infection patients and healthy controls to assess small molecule perturbations. Unfortunately, the methods and results sections are extremely vague, so it was difficult to understand the analyses performed and if they are valid. Therefore, as written this manuscript should not be accepted to mSphere. Please see below for my specific comments.

Revisions and Comments:

1. Results: The results section of the manuscript was extremely difficult to follow, and the results were not well described. Specifically, I was not sure how the metabolite features were identified and how ambiguous identifications were dealt with. Also, since many isomers exist for bile acids and fatty acids, how were these identifications made as they could be wrongly identified based on mass and retention times alone.

We have added the following section: ". Metabolite annotations had to meet 3 criteria: 1) a high-confidence mass match to either the M+H or M-H adduct, 2) membership in a significantly regulated metabolic pathway, and 3) correlation with other annotated metabolites in a significant metabolic pathway. When possible, identities were further supported as either medium- or high-confidence using xMSAnnotator, a multi-stage clustering algorithm to derive compound annotation and confidence scores" (Uppal et al. Anal Chem 2017). [Line 272-279]

2. Results: Details about the pathway analyses were not described.

The mummichog program was used for the pathway analysis. We have provided additional details about the program and commands used in the revised methods section to support reproducibility in response to this comment. [Line 288-301]

3. Results: Some of the statements in the results are not correct. For example, "Using a false discovery rate (FDR)-corrected p-value of 0.05..". Most people use p-values less than or equal to 0.05 in their statistical analyses. Additionally, the type of FDR correction was not listed or explained, and neither were the statical analyses performed on the data.

We used the Benjamini-Hochberg method for multiple hypothesis testing for the FDR calculation. This method was cited in the original manuscript, and we have added the named method to the revised methods section. [Line 285-287]

For '-omics' assays such as metabolomics that simultaneously measure thousands small molecules in biologic samples, it is generally advisable to use statistical methods that account for multiple hypothesis testing (rather than using $p < 0.05$).

4. Methods: Information on instrumental parameters is missing from the manuscript such

as LC gradients, flow rates, mobile phases, ionization source settings, mass spectrometry setting, MS resolving power, utilized targeted exclusion lists, etc. These experiments therefore cannot be understood or reproduced with all the missing details.

We have added additional details about the LC-MS method used for analysis, including LC gradients, flow rates, ionization source settings and mass spectrometry settings to the revised methods section. [Line 258-270]

Re: Spectrum02826-25R1 (High-resolution metabolomic analysis of stool reveals expanded biomarkers of *C. difficile* colitis and insights into pathophysiology)

Dear Dr. Nirja Mehta:

Thanks for addressing the final Spectrum-specific items in the submission system. Your paper has now been accepted for publication, congratulations!

Your manuscript has been accepted, and I am forwarding it to the ASM production staff for publication. Your paper will first be checked to make sure all elements meet the technical requirements. ASM staff will contact you if anything needs to be revised before copyediting and production can begin. Otherwise, you will be notified when your proofs are ready to be viewed.

Sincerely,
Jan Claesen
Editor
Microbiology Spectrum